# Exosomes derived from M2 Macrophages Improve Angiogenesis and Functional Recovery after Spinal Cord Injury through HIF-1α/VEGF Axis

**DOI:** 10.3390/brainsci12101322

**Published:** 2022-09-29

**Authors:** Jiang-Hu Huang, Hang He, Yong-Neng Chen, Zhen Liu, Manini Daudi Romani, Zhao-Yi Xu, Yang Xu, Fei-Yue Lin

**Affiliations:** 1Department of Orthopedics, Fujian Provincial Hospital, Fujian Medical University, Fuzhou, 350001, China; 2Department of Social Economy and Business Administration, Woosuk University, Wanju-gun 55338, Korea; 3Department of Spine Surgery and Orthopaedics, Xiangya Hospital, Central South University, Changsha 410008, China

**Keywords:** M2 macrophage-derived exosomes, angiogenesis, HIF-1α, spinal cord injury, functional recovery

## Abstract

Exosomes are nano-sized vesicles that contain a variety of mRNAs, miRNAs, and proteins. They are capable of being released by a variety of cells and are essential for cell–cell communication. The exosomes produced by cells have shown protective benefits against spinal cord damage (SCI). Recently, it was discovered that M2 macrophages aid in the angiogenesis of numerous illnesses. However, the functional role of M2 macrophage-derived exosomes on SCI is unclear. Here, we investigate the pro-angiogenesis of M2 macrophage-derived exosomes on SCI. We founded that M2 macrophage exosomes alleviated tissue damage and enhanced functional recovery post-SCI. We discovered that M2 macrophage exosome administration increased angiogenesis after SCI in vivo using immunohistochemistry, immunofluorescence labeling, and Western blot analysis. Additionally, the expression of the pro-angiogenesis factors, HIF-1α and VEGF, were enhanced with the treatment of the M2 macrophage exosomes. Furthermore, we found that M2 macrophage exosomes enhanced neurogenesis after SCI in vivo. In vitro, we found that M2 macrophage exosomes can be taken up by the brain endothelial cell line (bEnd.3) and that they enhanced the tube formation, migration, and proliferation of bEnd.3 cells. Furthermore, by using special siRNA to inhibit HIF-1α expression, we observed that the expression of VEGF decreased, and the tube formation, migration, and proliferation of bEnd.3 cells were attenuated with the treatment of HIF-1α-siRNA. In conclusion, our findings reveal that M2 macrophage exosomes improve neurological functional recovery and angiogenesis post-SCI, and this process is partially associated with the activation of the HIF-1/VEGF signaling pathway.

## 1. Introduction

Acute spinal cord injury (SCI) is a common type of disease associated with trauma, leading to the loss of sensation and movement and causing a heavy burden on both the patient and their family [1]. An epidemiological analysis of SCI found that there are currently around 180,000 cases of SCI worldwide, and the number of cases is continuing to rise. Numerous drugs and approaches are effective for the treatment of SCI in the animal models, but none has yet been demonstrated in human clinical trials [2].

The pathological mechanism of acute SCI is quite complicated and has not yet been fully investigated. It is currently well acknowledged that it can be primarily split into primary and secondary injuries. The mechanisms of secondary injury include vascular injury, inflammation, and apoptosis [3]. Among them, vascular injury is one of the major pathological changes. Ischemia-reperfusion injury, a microvascular decrease, and the destruction of the vessel network integrity are all examples of vascular injury. Vascular injury during SCI can cause angiogenesis and subsequently enhance tissue healing, but repair is restricted as a result of the local microenvironment’s inhibiting effect [4,5,6,7]. Numerous studies have reported that stimulating angiogenesis can aid in functional recovery following SC [8,9,10]. I. Therefore, understanding how to encourage angiogenesis is essential for SCI rehabilitation. 

Inflammation is also an important pathological change of secondary injury following SCI [3,11,12]. Among them, macrophages have a crucial role [13,14]. The M1- and M2-type macrophages are two different subtypes of macrophage. Pro-inflammatory substances secreted by the M1-type macrophages worsen spinal cord injury, whereas anti-inflammatory factors released by the M2-type macrophages promote the repair of damaged spinal cord tissue [13,14]. Recent research has demonstrated that M2 macrophages can promote angiogenesis [15,16]. M2 macrophages were demonstrated to cause a significant amount of angiogenesis in the live chick embryo model [15]. According to another study, the use of M2 macrophages enhanced the quantity of tubular development and endothelial cells [16]. However, the role of M2 macrophages in SCI still needs to be clarified. 

Exosomes are nano-sized vesicles (30–150 nm) that contain a variety of mRNAs, miRNAs, and proteins. They have been found in various cells and tissues and play a vital role in cell–cell communication [17]. There are several advantages of using exosomes to cure illnesses. Firstly, exosomes have a stable dual-layer membrane structure that is difficult to destroy. Secondly, because exosomes are smaller than cells and might not obstruct microvessels, they are safer. Exosomes are nanoscale vehicles that can pass through the plasma membrane, which is the third point [18,19]. They can reportedly cross the blood–brain barrier and are less likely to be rejected by the immune system. Thus, researchers have been paying more attention to the use of exosomes to treat neurological illnesses [20,21,22,23]. Currently, exosomes from LPS-stimulated macrophages have been found to have neuroprotective effects and to promote functional recovery in rats after ischemic stroke [24]. 

According to the above evidence, this study intends to study whether M2 macrophage-derived exosomes have a role in promoting angiogenesis and its underlying mechanism in SCI.

## 2. Materials and Methods

### 2.1. Induction and Identification of M2 Macrophages 

Adult male SD rats aged 4–6W were selected and sacrificed by being deeply anesthetized with isoflurane (1–3%) inhalation anesthesia. After taking out the femur and tibia, the syringe drew out the cells in the marrow cavity. The cell suspension was filtered with a sterile 70-mesh cell strainer; the filtrate was collected in a centrifuge tube. Unattached cells were collected via centrifugation (1200 rpm, 4 min); the supernatants were discarded; the cells were resuspended in a 30 ng/mL M-CSF solution cell culture medium and evenly planted in a Petri dish to stimulate differentiation of precursor monocytes into macrophages for 3–4 days. The adherent cells were M0-type macrophages. On days 6–7 of the M0 macrophage culture, the medium was removed and rinsed once with PBS, and then, M2 medium (20 ng/mL IL-4, 20 μg/mL gentamycin, 10% FBS-LE) was added. After 24 h, the M2-type macrophages were induced. The cells were transferred to a glass slide, and the cell morphology was observed using a conventional light microscope. The expression of the M2-type macrophage-specific markers CD206 and ARG were then detected by flow cytometry, Western blotting (WB), and immunofluorescence staining.

### 2.2. Flow Cytometric Analysis

Flow cytometric analysis was conducted to evaluate the characteristics of the M2 macrophages after labeling with ARG (eBiosciences, San Diego, CA, USA). The isotype matched to the mouse antibody (BD Biosciences) was used as a control. The M2 macrophages were processed with a FACSCalibur system (BD Biosciences).

### 2.3. The Extraction and Identification of Exosomes

The M2-type macrophage exosomes were collected as described previously [25]. After growing the M2 macrophages on the flask bottom (approximately 80%), the cells were rinsed thrice with PBS; the M2-type macrophage serum-free medium was refreshed, and the cells were cultured for 48 h. The cells were centrifuged (1000× *g*, 10 min, 4 °C) to eliminate residual cells and centrifuged again (2000× *g*, 20 min, 4 °C) to remove cell debris. After passing through a 0.22 μm filter, the supernatant was transferred to an ultrafiltration tube with a membrane pore size of 100,000 Daltons and centrifuged (3220× *g*, 4 °C). Then, the concentrate was placed in an ultracentrifuge tube and centrifuged (10,000× *g*, 2 h, 4 °C) to derive the M2 macrophage exosomes. The supernatant was carefully removed; PBS was added; it was mixed and centrifuged again; this process was repeated 3 times to remove residual protein. After removing the supernatant, the M2 macrophage exosomes were dissolved with 500 μL PBS and placed in a sterile EP tube for further analysis. WB, nanoparticle tracking analysis (NTA), and transmission electron microscopy (TEM) were employed to identify exosomes. 

### 2.4. Construction of a Contusion SCI Rat Model

Adult Sprague–Dawley rats (180–220 g, male) were obtained from the Animal Center of Fujian Provincial Hospital (Fujian, China). The experimental protocols are in compliance with the Ethics Committee of Fujian Provincial Hospital (approval number: K2017-02-018). All rats were maintained in individual cages, with controlled light cycle and temperature and unlimited access to water and food.

The SCI rat model was constructed as described previously [25]. The rats were deeply anesthetized with isoflurane (1–3%) inhalation anesthesia, and then, the rat skin was routinely prepared, disinfected, and paved. The T10 spinous process was fixed as the center; the skin was cut in the median line; and the muscle was eliminated to expose the lamina and T9-T11 spinous process. The lamina and T10 spinous process was bitten, resulting in a ~3 mm circular area centered on the SC segment of T10. A striking rod (8 g) was utilized to fall from the T10 SC at 4 cm in height and centered on the median arterial vessels at the back of the SC. With the intramedullary hemorrhage in the SC, the body and the lower limbs fluttered and retracted, and the rat tails swayed and were observed by the naked eye. The wound was rinsed, closed layer by layer, and covered. After disinfection with complex iodine once daily for three days, the hind limbs of the rats were alternately injected with 4WU penicillin daily for three days. The rats in the sham group received laminectomy only.

### 2.5. Experimental Design 

Sixty animals were randomly assigned to 3 groups (*n* = 20/group). In the sham group, the the animals underwent laminectomy only; in the SCI + saline group, the animals were exposed to SCI and then treated with 0.5 mL saline via tail vein injection after 30 min of SCI; in the SCI + exosomes group, the animals were exposed to SCI and then treated with M2 macrophage exosomes (100 μg) in 0.5 mL saline via tail vein injection after 30 min of SCI. At the scheduled experimental time points, the rats were deeply anesthetized with isoflurane (1–3%) inhalation anesthesia, and the rats were sacrificed. The spinal cord tissues (10 mm), including the lesion center, were collected for subsequent experiments.

### 2.6. Behavioral Evaluation

The locomotor recovery of the SCI rats was determined with the Basso, Beattie, and Bresnahan (BBB) scale [26]. Behavioral evaluation was conducted at 1 and 3 days and at 1, 2, 3, and 4 weeks post-SCI. The movement of animals was examined by two well-trained and independent observers who were blinded to the treatments, and the scores were measured based on the BBB scale. The movement of each rat was recorded in 5 min and repeated thrice. The mean value was then calculated. Lastly, the mean score of the two evaluators was determined.

### 2.7. Identification of Lesions via Cresyl Violet Staining

After 28 days of SCI and behavioral evaluation, 10 mg/kg chloral hydrate (10%) was used to sacrifice the rats. The SC (10 mm long), including the lesion center, was obtained for cresyl violet staining. The severest injury transverse sections were identified as the lesion epicenter. Cresyl violet acetate (0.5%) was stained on every 40th section of the lesion sites and then observed under a BH-2 microscope (Olympus, Melville, NY, USA). The lesion site was outlined and quantified with Image-Pro Plus 6.0 software (Media Cybernetics, Rockville, MD, USA). Transverse slides, with an interval of 400 μm from the rostral to caudal areas, were assessed up to 800 μm from the lesion epicenter. The ratio presenting as the ‘‘lesion area/total area’’ of the slides in each group was examined.

### 2.8. Immunofluorescence Staining

To assess the angiogenic effect of the M2 macrophage exosomes on the SC of the rats post-SCI, the proportion of proliferating blood vessels was detected by immunofluorescence staining. After SCI for 3 days, the SC tissues were collected and fixed with paraformaldehyde (4%) for 3 h. After incubation with 6% sucrose in PBS overnight, an optimal cutting temperature compound (Sakura Finetek, Torrance, CA, USA) was used to embed the tissues and then sectioned at 5 μm thickness (*n* = 5/group). After air-drying for 15 min, 10% normal goat serum/PBS was added to the sections for 1 h, and then, they were exposed to proliferating cell nuclear antigen (PCNA, Shanghai, China) rabbit monoclonal antibody (1:200; Abcam, Cambridge, UK) or CD31 mouse monoclonal antibody (1:100; Abcam, Cambridge, UK), Nestin (1:200, Abcam, Cambridge, UK), NeuN (1:200, Abcam, Cambridge, UK), and Sox2 (1:200, Sangon Biotech, Shanghai, China) overnight at 4 °C. After washing with PBS, the sections were exposed to the secondary antibody Cy3 goat anti-mouse IgG (H + L) (1:1000, Jackson, Bar Harbor, ME, USA) or Alexa Fluor 488 goat anti-rabbit IgG (H + L) (1:1000, Jackson, Bar Harbor, ME, USA) for 30 min. The sections were rinsed thrice with PBS and exposed to DAPI solution (CST, Boston, MA, USA) for 15 min. The number of positive cells in the SC was counted.

### 2.9. Immunohistochemistry Analysis 

The paraffin sections (4 µm thickness) were cut transversely via the lesion area. After deparaffinizing and antigen retrieving, the section was exposed to primary antibody against CD31 (1:200, Abcam, Cambridge, UK) overnight at 4 °C. Next, the section was exposed to secondary antibodies (anti-rabbit IgG antibodies) for 1 h. Lastly, the section was stained with diaminobenzidine (DAB) for visualization and was recorded with optical microscopy (Olympus). The Image-Pro Plus 6.0 was employed to count the positive cells.

### 2.10. bEnd.3 Cell Culture and Oxygen Glucose Deprivation (OGD)

The mouse bEnd.3 cells were supplied by the Type Culture Collection of Chinese Academy of Sciences (Shanghai, China). The cells were grown in DMEM (Sigma, MO, USA) containing 1% penicillin-streptomycin, 10% FBS and 4500 mg glucose/L at 37 °C and 5% CO_2_. Around 50% of the medium was replaced every three days.

After incubated with PBS, the exosomes, the exosomes + HIF-1α-siRNA, or the exosomes + Con siRNA for 6 h, the cells were exposed to OGD. The medium was changed with glucose-free DMEM (Gibco, NY, USA) and then incubated in a humidified and anaerobic incubator, which was infused with O_2_ (1%), CO_2_ (5%), and N_2_ (94%) at 37 °C for 4 h. After rinsing twice with RPMI 1640, the cells were placed in a normal culture medium for 6 h.

### 2.11. Uptake of Exosomes by bEnd.3 Cells

The exosomes were labeled with a green fluorescent dye (PKH67, Sigma), as described previously [27]. The exosomes were co-cultured with bEnd.3 cells at 37 °C for 3 h, followed by fixing in 4% PFA for 15 min. After rinsing thrice with PBS, the nuclei were stained with DAPI (1:500, Invitrogen, CA, USA) for 5 min. The signals were determined using a fluorescence microscope.

### 2.12. EdU test

An EdU test was conducted to explore the effect of the M2 macrophage exosomes on bEnd.3 cell proliferation. Briefly, 1 × 10^5^ cells were grown in 10% FBS-containing DMEM for 24 h, followed by incubation for 2 h with EdU-labeling reagent (1:1000, Invitrogen). The medium was discarded and washed twice with PBS for 5 min. After incubation with 4% PFA for 30 min and glycine (2 mg/mL) for 5 min, the cells were rinsed with PBS for 5 min and determined with the Click-iT Edu Alexa Fluor 555 Imaging Kit (Invitrogen). After staining with DAPI (1:500, Invitrogen) for 5 min, the nuclei were examined using a fluorescence microscope, and the numbers of EdU+ cells were counted.

### 2.13. Capillary Network Formation (CNA) Assay

After treatment with PBS, exosomes, exosomes + HIF-1α-siRNA, or exosomes + Con siRNA for 4 h, the bEnd.3 cells (1 × 10^5^/well) were exposed to OGD for 4 h and cultured in a 96-well plate containing Matrigel. After incubation for 6 h, the tube formation was then determined using an inverted microscope. The Image-Pro Plus 6.0 was employed to assess the network structure.

### 2.14. Transwell Assay

After pretreatment with PBS, exosomes, exosomes + HIF-1α-siRNA, or exosomes + Con siRNA, the bEnd.3 cells were subjected to OGD for 4 h. The HUVECs cells were grown in serum-free medium in the top chamber containing Matrigel (BD, USA). The bottom of the Transwell^®^ (Corning, Guangdong, China) plates was filled with the migration-inducing medium, while 1 × 10^5^ cells were grown in the top chamber. After 24 h, the residual cells in the top chamber were eliminated, and the migrated cells were subjected to 0.1% crystal violet staining. An Olympus inverted microscope was employed to count the numbers of cells.

### 2.15. siRNAs

To assess the effects of HIF-1α on M2 macrophage exosome-induced VEGF expression and angiogenesis on bEnd.3 cells, three HIF-1α-specific siRNAs (si HIF-1α #1, 2 and 3; RioboBio, Guangzhou, China) were applied to decrease the expression of HIF-1α in the bEnd.3 cells. The bEnd.3 cells were transfected with siHIF-1α or negative control (Con) siRNA by the Lipofectamine 2000 (Invitrogen). The inhibitory efficiencies of these siRNAs were evaluated by qRT-PCR 24 h later. Then, the most efficient siRNA was selected for the subsequent experiments.

### 2.16. qRT-PCR

TRIzol (Invitrogen) was applied to extract total RNA from the bEnd.3 cells. cDNA synthesis was conducted with the PrimeScript RT reagent Kit (TaKaRa, Japan). Additionally, qRT-PCR was conducted on an ABI PRISM^®^ 7900HT System (Applied Biosystems, USA) using the SYBR Premix Ex Taq (TaKaRa). The delta-delta CT method was employed to analyze gene expression. GAPDH was employed as a reference control. The following primer pairs were applied: 

HIF-1α: 

F: 5′-TGAACAGGATGGAATGGAGCA-3′, 

R: 5′-GTGATCTGGCATTCGTAAGG-3′); 

VEGF: 

F: 5′ ATCATGCGGATCAAACCTCACC 3′, 

R: 5′ GGTCTGCATTCACATCTGCTATGC 3′); 

GAPDH: 

F: 5′-TGGGCTACACTGAGCACCAG-3′, 

R: 5′-AAGTGGTCGTTGAGGGCAAT-3′.

### 2.17. WB Analysis

Total protein was extracted from the cells or SC tissues, and its content was determined using a BCA protein assay kit (Beyotime, Shanghai, China). After separation through a 10% SDS/PAGE, the protein samples were transferred onto a PVDF membrane (Millipore, USA). After inhibiting with 5% skimmed milk for 1 h, the membrane was exposed to primary rabbit polyclonal anti-CD63 (1:1000; Abcam), anti-CD9 (1:1000; Abcam), anti-TSG101 (1:1000; Abcam), anti-ARG (1:500; Abcam), anti-CD206 (1:1000; Abcam), anti-HIF-1α (1:1000, Abcam), and anti-VEGF antibody (1:200, Abcam) at 4 °C overnight. After rinsing with PBS, the membrane was exposed to anti-mouse IgG HRP-conjugated secondary antibody (1:2000; Jackson) for 1 h. Lastly, an ECL kit (Beyotime, China) was employed to visualize the blots. The GAPDH was employed as a reference control.

### 2.18. Statistical Analysis

All statistical tests were conducted with SPSS v17.0 (SPSS, Inc., Chicago, IL, USA). The values were shown as mean ± standard deviation (SD). The BBB scores were compared with repeated measures analysis of variance (ANOVA), followed by the Bonferroni post hoc corrections. One-way ANOVA was employed to compare the means among the multiple groups. In addition, the mean between 2 groups was compared with an independent *t*-test. In addition, *p* < 0.05 was deemed statistically significant.

## 3. Results

### 3.1. Characterization of M2 Macrophage Exosomes

WB, flow cytometry, and immunofluorescence staining were conducted to analyze the surface markers of the M2 macrophages. The results demonstrated that CD206 and ARG were expressed in these cells, and flow cytometry showed that 91.2% of the cells were ARG+ cells (Figure 1A). The WB, NTA, and TEM were performed to analyze the exosomes. Based on the results of the TEM, the exosomes had a biconcave hemispherical structure of approximately 100 nm (Figure 1B). The NTA revealed that the range of exosome sizes was 30–200 nm and peaked at 85 nm (Figure 1C). The exosome specific markers (e.g., TSG101, CD63, and CD9) were further verified by WB (Figure 1D).

### 3.2. M2 Macrophage Exosomes Attenuate Tissue Damage (TD) and Improve Functional Recovery (FR) Post-SCI

Cresyl violet staining was conducted to assess the protective effect of the M2 macrophage exosomes after SCI for 28 days. The thoracic SC sections were evaluated, and the ratio of ‘‘lesion area/total area’’ was detected for each section (Figure 2A). The lesions were greater in the saline-treated rats, not only at the injury epicenter (Figure 2B) but also at the areas caudal and rostral to the epicenter. The M2 macrophage exosomes significantly reduced the lesion volume (Figure 2A,B). The BBB scale was employed to assess the FR post-SCI. It was found that the scores of the two groups were 0 after SCI, which indicated that the SCI rat model was constructed successfully. The hind limb function of the SCI rats was spontaneously recovered, and the BBB scores were gradually increasing in both groups (Figure 2C). After SCI for 14 days, the treatment with the M2 macrophage exosomes exhibited higher scores when compared to the PBS treatment group (days 14, 21, and 28; *p* < 0.01) (Figure 2C).

### 3.3. M2 Macrophage Exosomes Promote Angiogenesis after SCI

CD31 immunohistochemistry analysis was conducted to assess the existence of blood vessels in the SC after SCI for 3 days. The data indicated that the number of blood vessels in the saline treatment group was remarkably decreased compared to the sham group (*p* < 0.01; Figure 3A,C). Moreover, the M2 macrophage exosomes markedly increased the numbers of blood vessels compared to the saline group (*p* < 0.01; Figure 3A,C). In addition, the M2 macrophage exosomes could increase the protein expression level of HIF-1α and VEGF as compared to the saline group (*p* < 0.01; Figure 3B,D,E)). Subsequently, the number of proliferating blood vessels was determined by immunofluorescence staining. The numbers of PCNA/CD31 positive cells were higher in the saline treatment group than in the sham group (*p* < 0.05; Figure 4A,B). This indicates that angiogenesis occurs in the SC after SCI. Moreover, the M2 macrophage exosome group had a significantly higher number of PCNA/CD31 positive cells than the saline group (*p* < 0.01; Figure 4A,B).

### 3.4. M2 Macrophage Exosomes Promote Neurogenesis after SCI

To determine whether the M2 macrophage exosomes promote neurogenesis, the immunofluorescence staining was conducted to assess the expression of Nestin, Sox2, and NeuN. The numbers of Nestin-, Sox2-, and NeuN+ cells in the SC were remarkably higher in the M2 macrophage exosome treatment group than in the saline group (*p* < 0.01; Figure 5).

### 3.5. The M2 Macrophage Exosomes Exhibit Pro-Angiogenic Effects on bEnd.3 Cells

To explore whether the M2 macrophage exosomes can be taken up by the bEnd.3 cells, PKH67 was employed to label the M2 macrophage exosomes. Next, the labelled exosomes were added to the bEnd.3 cells. It was found that PKH67 localized in the cytoplasm of the bEnd.3 cells (Figure 6A). These findings reveal that the M2 macrophage exosomes could be taken up by the bEnd.3 cells.

To examine the effect of the M2 macrophage exosomes on the bEnd.3 cell proliferation, the EdU test was carried out. The proportion of EdU+ cells was markedly higher in the M2 macrophage exosome treatment group than in the PBS group (*p* < 0.01; Figure 6B,C).

Both the CNA and the transwell assays were performed to determine the pro-angiogenic effects of the M2 macrophage exosomes on the bEnd.3 cells. The CNA assay revealed that the total tube length and total branch point numbers were higher in the M2 macrophage exosome group than in the PBS group (*p* < 0.01; Figure 6D–F). Furthermore, the migrated bEnd.3.cells were dramatically increased in the M2 macrophage exosome group compared to the PBS group (*p* < 0.01; Figure 6G,H).

### 3.6. HIF-1α Is Required in M2 Macrophage Exosomes to Promote VEGF Expression and ANGIOGENESIS on bEnd.3 cells

It has been proved that HIF-1α promotes angiogenesis by regulating its downstream target gene VEGF [28]. To explore the effect of HIF-1α in M2 macrophage exosome-induced VEGF expression and angiogenesis on bEnd.3 cells, HIF-1α expression was inhibited by three HIF-1α-specific siRNAs (#1, #2, and #3). The data indicated that the HIF-1α expression was greatly repressed by HIF-1α-siRNA#1, #2, and #3 and that HIF-1α-siRNA#1 was the most effective siRNA. Then, HIF-1α-siRNA#1 was selected for the subsequent analyses (*p* < 0.01; Figure 7A). Meanwhile, the VEGF expression was dramatically reduced by HIF-1α-siRNA (P<0.01; Figure 7B,C). The proportion of EdU+ cells was obviously lower in the HIF-1α-siRNA group than in the scramble-siRNA group (*p* < 0.01; Figure 7D,E). The total tube length and total branch point number were also markedly lower in the HIF-1α-siRNA group than in the scramble-siRNA group (*p* < 0.01; Figure 7F–H). Additionally, the migrated bEnd.3 cells were remarkably decreased in the HIF-1α-siRNA group compared to the scramble-siRNA group (*p* < 0.01; Figure 7I,J).

## 4. Discussion

Exosomes are double-encapsulated structures with a diameter of about 100 nm, containing various cargo, including mRNAs, miRNAs, and protein. They perform their function by transferring these molecules to their target cells [17,19]. To our knowledge, this study is the first to assess the effect of M2 macrophage exosomes on angiogenesis post-SCI. Our results demonstrated that treatment with M2 macrophage exosomes significantly attenuated tissue damage and enhanced functional recovery post-SCI. In addition, the M2 macrophage-Exo treatment promoted angiogenesis in vivo. Additionally, in vitro, our results showed that the M2 macrophage-derived exosomes improved the tube formation, migration, and proliferation of endothelial cells. Furthermore, we found that the role and mechanism of these exosomes on angiogenesis may partly be by the regulating HIF-1α/VEGF pathway.

Exosomes offer a number of benefits, including their tiny size, stability, and resistance to degradation. They can also cross the blood–brain barrier, etc. [16,21]. The researchers’ interest in using exosomes to treat SCI has recently increased. The exosomes produced by several cells exhibited a protective effect against SCI [25,29,30]. The exosomes secreted by hucMSCs have a protective role after SCI by attenuating inflammation [29]. In another study, the researchers demonstrated that BMSCs-Exo treatment promoted functional recovery after SCI by deactivating the A1 neurotoxic reactive astrocytes [28]. In our previous work, we also found that mesenchymal stem cell-derived exosomes decreased inflammation and apoptosis and promoted angiogenesis post-SCI [25]. In this study, we successfully extracted exosomes from the M2 macrophage. We also found that the M2 macrophage exosomes could alleviate tissue damage and enhance functional recovery after SCI.

After SCI, the integrity of the spinal cord vascular network is destroyed, and the vascular endothelial cells are lost, which leads to spinal cord local ischemia, neuron cell ischemia, and apoptosis, and in turn, aggravates the spinal cord dysfunction [4,7]. Promoting angiogenesis is crucial for the reconstruction of spinal cord tissue after injury. In a variety of disease models, M2 macrophages have been shown to have the capability for promoting angiogenesis [15,16,31,32,33]. In the wound healing model, the M2 macrophages were shown to promote angiogenesis during the early phase [32]. They also revealed that M2 macrophages were mainly located in the perivascular areas in the wound healing model [33]. Using the axonal injury model, the depletion of CD11b+ cells significantly decreased blood vessel formation in the sciatic nerve distal stump [31]. In the live chick embryo model, Ewa et al. found that M2 macrophages induced a higher level of angiogenesis when compared to M0-/M1-macrophages [14]. Moreover, M2 macrophages injected subcutaneously increased tubular formation and the proliferation of endothelial cells in mice [15]. Exosomes have many advantages over the treatment of SCI with the application of cells. For example, they will not proliferate abnormally and can cross the blood–brain barrier, etc. Thus, this research aims to detect the effect of M2 macrophage exosomes on spinal cord injury after SCI. We found that the M2 macrophage exosome treatment significantly increases the tube formation, migration, and proliferation of endothelial cells in vitro and could promote the angiogenesis in the spinal cord injury model.

The mechanism of angiogenesis following SCI is complex. Many signaling pathways were proved to be involved in this process. Among them, HIF-1α/VEGF signal transduction plays an essential role in angiogenesis after SCI [33,34,35]. The expression of HIF-1α increased following SCI, peaked at day 3, and gradually decreased to basal levels [34]. After spinal cord injury, HIF-1α promotes angiogenesis by regulating its downstream target gene VEGF [36]. Here, we observed that the levels of HIF-1α and VEGF were upregulated at day 3 post-SCI, and the M2 macrophage exosome treatment significantly enhanced their expression in vivo. To investigate whether the HIF-1α/VEGF signaling pathway was involved in this process with the treatment of M2 macrophage exosomes, we used siRNAs to inhibit HIF-1α expression and found that the VEGF expression significantly decreased. Additionally, the promotion of angiogenesis with the treatment of M2 macrophage exosomes was also attenuated. These results indicated that the HIF-1α/VEGF signaling pathway is responsible for the angiogenesis process with the treatment of M2 macrophage exosomes. However, our study still has a limitation; we have not explored how the role of M2 macrophage exosomes affects the angiogenesis via the activation of the HIF-1α/VEGF signaling pathway after SCI in vivo. In a further study, we will continue to conduct experiments to confirm this underlying mechanism during this process.

Spinal cord endogenous neural stem cells (SCENSCs) play a vital role during SCI repair [35,37]. Under normal physiological conditions, the SCENSCs are in a relatively static state. When a certain pathological state occurs in the spinal cord or under the action of external cytokines, the SCENSCs are activated and proliferate and differentiate into neurons, thereby participating in the repair of the damaged spinal cord [38,39]. Nestin and Sox2 are two biomarkers of SCENSCs; in this study, we found that the Nestin positive cells and Sox2 positive cells increased after SCI; this was consistent with a previous study [38]. Moreover, the M2 macrophage exosome treatment significantly enhanced their positive cell number expression. Meanwhile, the M2 macrophage exosome treatment also increased the number of neurons after SCI. These findings demonstrate that the M2 macrophage exosome treatment could promote neurogenesis after SCI.

## 5. Conclusions

This study demonstrates that the M2 macrophage exosome treatment promotes angiogenesis and neurogenesis, alleviates tissue damage, and enhances functional recovery after SCI, and this process is partially associated with the activation of the HIF-1/VEGF signaling pathway, which could be used as a novel treatment strategy for SCI repair.

## Figures and Tables

**Figure 1 brainsci-12-01322-f001:**
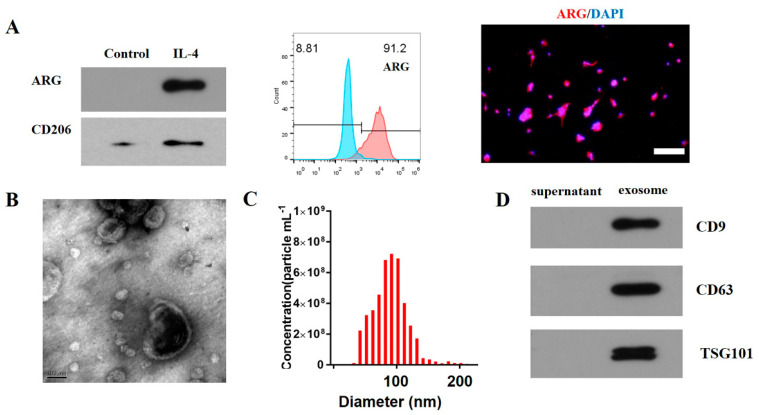
Characterization of M2 macrophages and exosomes. Western blot analysis of ARG and CD206 protein levels (**A**). Flow cytometry showed that 91.2% of cells were ARG+ cells (**A**). Immunofluorescence stain of the surfer markers—ARG (red) of M2 macrophages (**A**). The transmission electron micrograph of exosomes (**B**). The NTA of the exosomes (**C**). The exosome specific markers (e.g., TSG101, CD63, and CD9) were detected by WB (**D**). Scale bar = 100 µm (**A**). Scale bar = 100 nm (**B**).

**Figure 2 brainsci-12-01322-f002:**
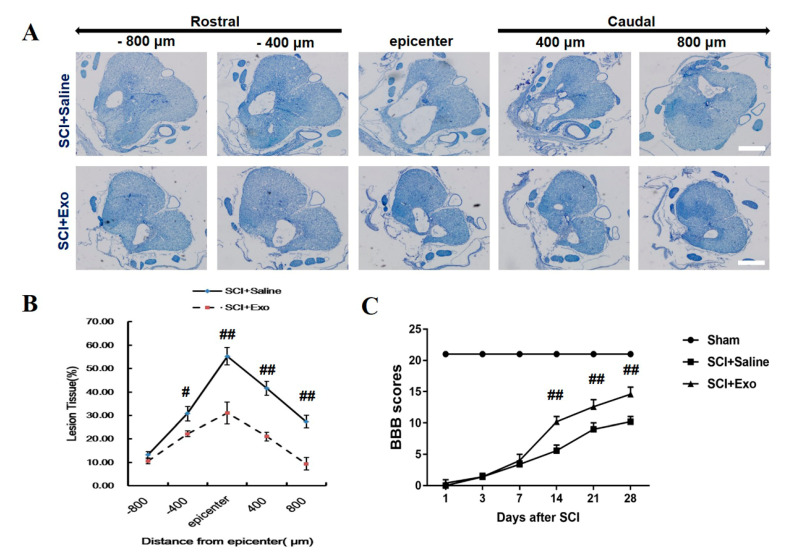
M2 macrophage exosomes attenuate TD and improve FR post-SCI. Representative cresyl violet staining of the SC lesion cavity (**A**). Quantification of the lesion cavity; the ratio presents as ‘‘lesion area/total area’’, 800 µm caudal and rostral to the epicenter, after 28 days of SCI (**B**). FR of rats was accessed by BBB scores and ranged from day 1 to day 28 (**C**). Mean ± SD. # *p* < 0.05 SCI + Exo versus PBS group, ## *p* < 0.01 SCI + Exo versus PBS group. *n* = 5 per group. Scale bar = 1 mm.

**Figure 3 brainsci-12-01322-f003:**
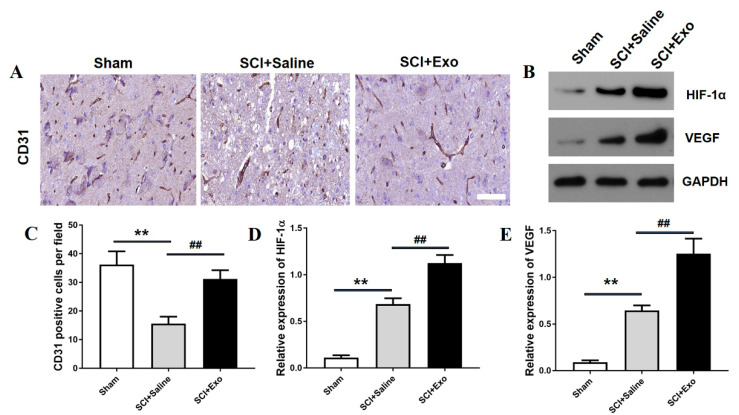
The administration of M2 macrophage exosomes improves angiogenesis post-SCI. The existence of blood vessels in the SC after SCI for 3 days was determined by CD31 (**A**). The number of the CD31+ cells (**C**). The protein levels of HIF-1α and VEGF were evaluated by WB. GAPDH expression was employed as an internal control (**B**,**D**,**E**). Mean ± SD. ** *p* < 0.01 SCI + saline group versus sham group, ## *p* < 0.01 SCI + Exo versus saline group. *N* = 5 per group. Scale bar = 50 µm.

**Figure 4 brainsci-12-01322-f004:**
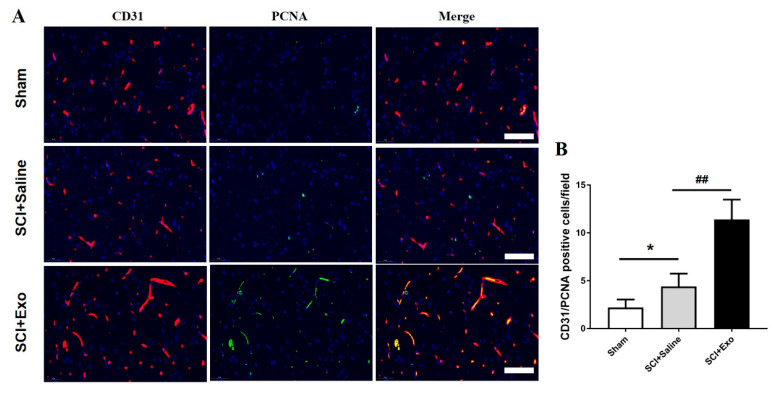
The administration of M2 macrophage exosome-induced endothelial cell proliferation after SCI in vivo. Representative images of PCNA+ (green) and CD31+ (red) cells in the anterior horn of the SC after 3 days of SCI (**A**). Quantification of PCNA/CD31+ cells in the injured SC (**B**). Mean ± SD. * *p* < 0.05 SCI + saline group versus sham group, ## *p* < 0.01 SCI + Exo versus saline group. *N* = 5 per group. Bar = 50 µm.

**Figure 5 brainsci-12-01322-f005:**
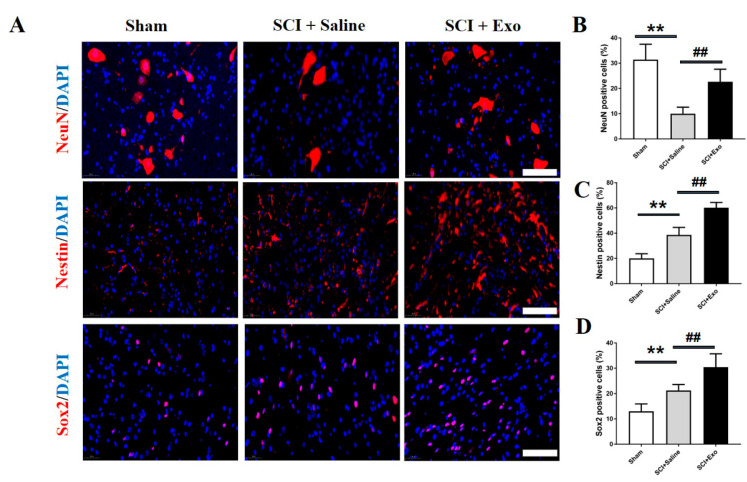
M2 macrophage exosomes promote neurogenesis post-SCI. Representative images of NeuN+ (red), Nestin+ (red), and Sox2+ (red) cells in the anterior horn of the injured SC (**A**). Quantification the NeuN+ cells, Nestin+ cells, and Sox2+ cells in the injured SC (**B**–**D**). Mean ± SD. ** *p* < 0.01 SCI + saline group versus sham group, ## *p* < 0.01 SCI + Exo versus saline group. *N* = 5 per group. Bar = 50 µm.

**Figure 6 brainsci-12-01322-f006:**
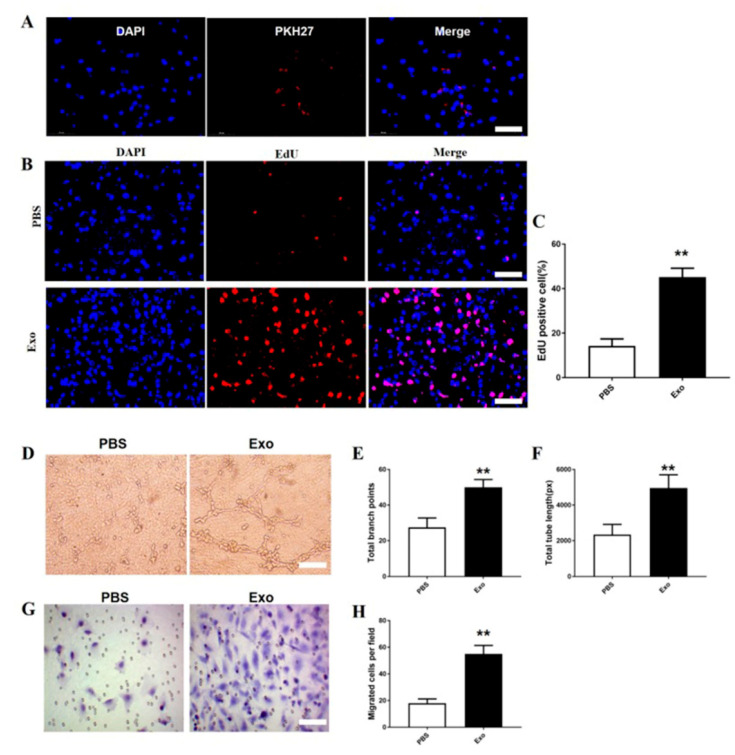
M2 macrophage exosomes attenuate angiogenesis, migration, and proliferation of bEnd.3 cells. PKH27-labelled exosomes uptake by bEnd.3 cells. Scale bar = 50 µm (A). The proliferation of HUVECs in each group (B). Quantification of EdU+ cells in each group (**C**). Tube formations were measured after growing bEnd.3 cells pre-treated with exosomes or PBS. Scale bar = 500 µm (D–F). Migration areas were measured in each group in bEnd.3 cells (**G**). Quantitative analysis of the numbers of the migrated bEnd.3 cells (**H**). Mean ± SD. ** *p* < 0.01 SCI + Exo versus PBS group. *N* = 5 per group.

**Figure 7 brainsci-12-01322-f007:**
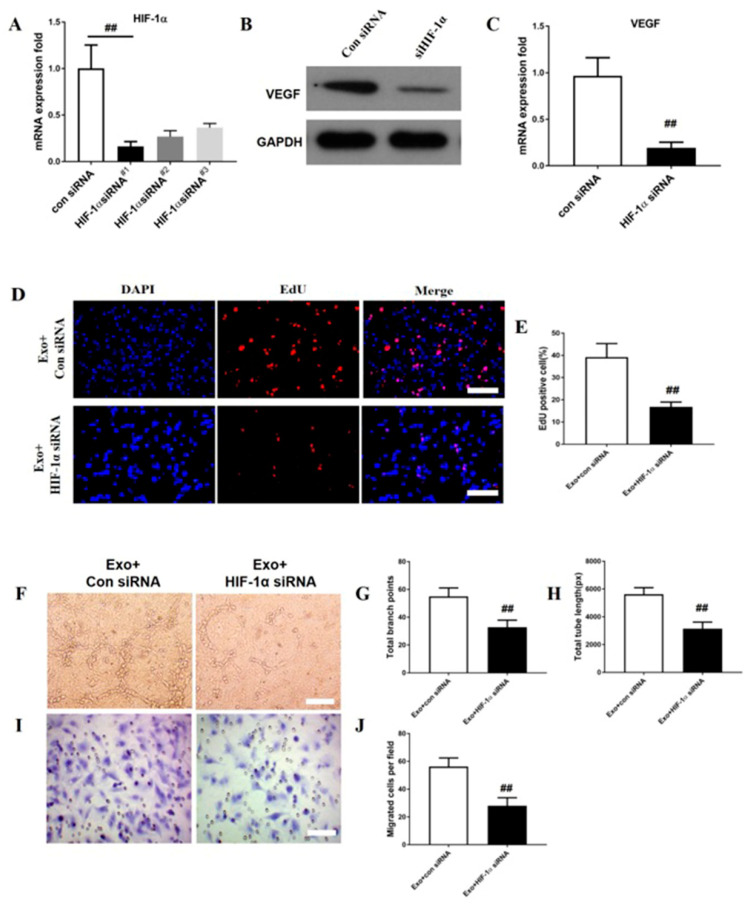
HIF-1α is required in M2 macrophage exosomes to promote VEGF expression and angiogenesis on bEnd.3 cells. qRT-PCR was conducted for analysis of the inhibitory efficiency of HIF-1α-siRNAs (#1, #2, and #3) (**A**). The protein levels of VEGF in HUVECs treated with M2 macrophage exosomes (**B**). The mRNA level of VEGF in bEnd.3 cells pretreated with M2 macrophage exosomes (**C**). The bEnd.3 cell proliferation test (D). Quantification of EdU+ cells in each group (**E**). Tube formations were assessed after growing bEnd.3 cells pretreated with exosomes or PBS. Scale bar = 500 µm (F–I). Migrated areas were measured in each group on bEnd.3 cells (**I**). Quantification of the numbers of the migrated bEnd.3 cells (**J**). Mean ± SD. ## *p* < 0.01 HIF-1α-siRNA group versus Con siRNA group. *N* = 5 per group.

## Data Availability

The raw/processed data required to reproduce these findings cannot be shared at this time as the data also form part of an ongoing study.

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
