# Peer review of "Exosomes derived from M2 Macrophages Improve Angiogenesis and Functional Recovery after Spinal Cord Injury through HIF-1α/VEGF Axis"

_brainsci, 2022, doi:10.3390/brainsci12101322_

Round 1
Reviewer 1 Report
The authors investigated the actions of M2 macrophage exosomes in a spinal cord contusion rat model. They found that M2 macrophage exosome treatment mitigated tissue damage, enhanced locomotor function recovery, and promoted neurogenesis after spinal cord injury. Moreover, their in vitro investigations of endothelial cells showed that M2 macrophage exosome treatment improved angiogenesis. Furthermore, the authors found that the actions of M2 macrophage exosome activity might be partly mediated by regulating the HIF-1alpha/VEGF pathway.
The topic addressed in this study is interesting and worth investigating. Such a study may contribute to our understanding of the molecular and neuronal mechanisms of the pathophysiology of spinal cord injury and provide clues for the development of new treatments for functional neurological disorders caused by spinal cord injury. However, some points need clarification, and certain statements require further justification.
1. Lines 110–121: "After anesthesia with 3 mg/kg chloral hydrate (10%), the rat skin was routinely prepared, disinfected and paved. [...]. The rats in sham group received laminectomy only. "
Chloral hydrate is a hypnotic rather than an anesthetic. The authors should explain why 10% chloral hydrate was used for animal surgery.
2. Lines 126–128: "then treated with M2 macrophage exosomes (100 μg) in 0.5 mL Saline via tail vein injection after 30 min of SCI."
Please explain why M2 macrophage exosomes were injected 30 min after spinal cord injury. If the injection of exosomes had been performed after more than 30 min, how do the authors believe this would have affected functional recovery?
3. Lines 132–134: "The movement of animals was examined by two well-trained and independent observers, and the scores were measured based on the BBB scale."
Were the observers blinded to the treatments?
4. "BBB" stands for "blood-brain barrier" in lines 63, 373, and 396.
It seems that "BBB" stands for "Basso, Beattie, and Bresnahan" in lines 131, 134, 239, 267, and 270.
An initialism used in a text should not stand for two different terms.
5. It seems that Figure 3D and E is not cited in the text.
6. Please explain what the blue spots indicate in Figure 5A.
7. The number of NeuN positive cells decreased in SCI+Saline group (Fig. 5B). Those of Nestin positive cells and Sox2 positive cells increased in SCI+Saline group (Fig. 5C and D).
Please explain why NeuN-positive cells decreased while nestin- and Sox2-positive cells increased after spinal cord injury.
8. Figure 6 and 7 legends: I had a hard time reading these legends. Parts of them should be described in the Materials and Methods section.
9. Figure 6 caption: "M2 macrophages exosomes attenuate angiogenesis, migration and proliferation of bEnd.3 cells."
Please check to ensure that this caption is correct.
10. Although I am not a native speaker of English, I noticed several grammatical errors, and it seems that there are several awkward expressions in the manuscript.
Author Response
The authors investigated the actions of M2 macrophage exosomes in a spinal cord contusion rat model. They found that M2 macrophage exosome treatment mitigated tissue damage, enhanced locomotor function recovery, and promoted neurogenesis after spinal cord injury. Moreover, their in vitro investigations of endothelial cells showed that M2 macrophage exosome treatment improved angiogenesis. Furthermore, the authors found that the actions of M2 macrophage exosome activity might be partly mediated by regulating the HIF-1alpha/VEGF pathway.
The topic addressed in this study is interesting and worth investigating. Such a study may contribute to our understanding of the molecular and neuronal mechanisms of the pathophysiology of spinal cord injury and provide clues for the development of new treatments for functional neurological disorders caused by spinal cord injury. However, some points need clarification, and certain statements require further justification.
Response:
We would like to thank both reviewers for their positive, insightful, and constructive comments regarding our manuscript. We have addressed all of the questions and concerns brought forth through additional experimentation and clarification. The following responses have been prepared to address all of the reviewers’ comments in a point-by-point fashion.
1. Lines 110–121: "After anesthesia with 3 mg/kg chloral hydrate (10%), the rat skin was routinely prepared, disinfected and paved. [...]. The rats in sham group received laminectomy only. " Chloral hydrate is a hypnotic rather than an anesthetic. The authors should explain why 10% chloral hydrate was used for animal surgery.
Response:We make a errors in this study, we check our experiment procedure and make sure the rat were deeply anesthetized with isoflurane (1–3%) inhalation anesthesia. We corrected this errors in the revised manuscript.
2. Lines 126–128: "then treated with M2 macrophage exosomes (100 μg) in 0.5 mL Saline via tail vein injection after 30 min of SCI." Please explain why M2 macrophage exosomes were injected 30 min after spinal cord injury. If the injection of exosomes had been performed after more than 30 min, how do the authors believe this would have affected functional recovery?
Response: We conducted this experiment according to the literature, as research published by Zhong et al, they reported that exosome were injected into the tail vein of each mouse, respectively, at 30 min after SCI. The time we performed exosome administrated my have affected functional recovery due to the pathophysiological change after SCI.
3. Lines 132–134: "The movement of animals was examined by two well-trained and independent observers, and the scores were measured based on the BBB scale." Were the observers blinded to the treatments?
Response: The Observers were blind to the treatments.
4. "BBB" stands for "blood-brain barrier" in lines 63, 373, and 396. It seems that "BBB" stands for "Basso, Beattie, and Bresnahan" in lines 131, 134, 239, 267, and 270. An initialism used in a text should not stand for two different terms.
Response: We used BBB for Basso, Beattie, and Bresnahan. Other items we used the full name blood-brain barrier" in lines 63, 373, and 396 to avoid confusion.
5.It seems that Figure 3D and E is not cited in the text.
Response: We corrected it.
6. Please explain what the blue spots indicate in Figure 5A.
Response: We added the figure label in Figure 5A.
7. The number of NeuN positive cells decreased in SCI+Saline group (Fig. 5B). Those of Nestin positive cells and Sox2 positive cells increased in SCI+Saline group (Fig. 5C and D). Please explain why NeuN-positive cells decreased while nestin- and Sox2-positive cells increased after spinal cord injury.
Response: NeuN positive cells represent Mature neuron. nestin- and Sox2-positive cells represent Neural Stem Cell Marker. The trauma may lead neuron damage and cause lost of NeuN-positive cells. Recent observations that resident nestin- and Sox2-positive cells express stem cell markers in ependyma ventriculorum areas of spinal cord. Spinal cord trauma may initiate the endogenous regenerative repair process and up regulated the nestin- and Sox2-positive cells and increased the cell numbers.
8. Figure 6 and 7 legends: I had a hard time reading these legends. Parts of them should be described in the Materials and Methods section.
Response: We revised our figure legends according to your suggestion and make it more clear to follow.
9. Figure 6 caption: "M2 macrophages exosomes attenuate angiogenesis, migration and proliferation of bEnd.3 cells." Please check to ensure that this caption is correct.
Response: It is correct.
10. Although I am not a native speaker of English, I noticed several grammatical errors, and it seems that there are several awkward expressions in the manuscript.
Response:We asked a native speaker to correct the grammatical errors through our manuscript.
Reviewer 2 Report
Well presented manuscript, centered on the recovery mechanisms that are implicated after SCI.
I would like to mention that this is a study that is based on mices and on in vitro experiments. This should be noticed in the title of the manuscript, in order to fully elucidate the content of the study. Apart from that, although studies that are based on experimental models and in vitro studies can offer useful preliminary results, they should be treated with caution. This should be more clearly delineated in the conclusion section of your manuscript.
Author Response
Response: We would like to thank both reviewers for their positive, insightful, and constructive comments regarding our manuscript.
Reviewer 3 Report
The original article by Huang et al. "Exosomes derived from M2 Macrophages Improve Angiogenesis and Functional Recovery after Spinal Cord Injury through HIF-1α/VEGF axi" covers a potentially interesting
and emerging topic related to the therapy of spinal cord injury. In this sense, this remains to be potentially interesting for the Brain Sciences readers.
I regard themain point of this paper as highly attractive as well as the results are clearly presented. The text does not contain any major errors, therefore I have some minor comments and recommendations:
1. There is a need to provide slightly more expanded introduction shortly
mentioning/describing statistics of SCI and its impact of modern healthcare.
2. The conclusion section should be improved
3. Following references should be added and properly cited within the main text:
- Tykocki T, Poniatowski ŁA, Czyz M, Wynne-Jones G. Oblique corpectomy in the cervical spine. Spinal Cord. 2018 May;56(5):426-435. doi: 10.1038/s41393-017-0008-4.
- Long HQ, Li GS, Cheng X, Xu JH, Li FB. Role of hypoxia-induced VEGF in blood-spinal cord barrier disruption in chronic spinal cord injury. Chin J Traumatol. 2015;18(5):293-5. doi: 10.1016/j.cjtee.2015.08.004. PMID: 26777714.
- Kubaszewski Ł, Wojdasiewicz P, Rożek M, Słowińska IE, Romanowska-Próchnicka K, Słowiński R, Poniatowski ŁA, Gasik R. Syndromes with chronic non-bacterial osteomyelitis in the spine. Reumatologia. 2015;53(6):328-36. doi: 10.5114/reum.2015.57639. Epub 2016 Feb 11. PMID: 27407266; PMCID: PMC4847283.
4. In some places the use of English could be improved on.
Completing this gaps will have an impact on the understanding the aim of the study and, from my point of view, is absolutely necessary.
Author Response
1. There is a need to provide slightly more expanded introduction shortly
mentioning/describing statistics of SCI and its impact of modern healthcare.
Response: We added some background about the statistics of SCI and its impact of modern healthcare in the introduction parts.
2. The conclusion section should be improved
Response:We revised the conclusion parts in our manuscript.
3. Following references should be added and properly cited within the main text:
- Tykocki T, Poniatowski ŁA, Czyz M, Wynne-Jones G. Oblique corpectomy in the cervical spine. Spinal Cord. 2018 May;56(5):426-435. doi: 10.1038/s41393-017-0008-4.
- Long HQ, Li GS, Cheng X, Xu JH, Li FB. Role of hypoxia-induced VEGF in blood-spinal cord barrier disruption in chronic spinal cord injury. Chin J Traumatol. 2015;18(5):293-5. doi: 10.1016/j.cjtee.2015.08.004. PMID: 26777714.
- Kubaszewski Ł, Wojdasiewicz P, Rożek M, Słowińska IE, Romanowska-Próchnicka K, Słowiński R, Poniatowski ŁA, Gasik R. Syndromes with chronic non-bacterial osteomyelitis in the spine. Reumatologia. 2015;53(6):328-36. doi: 10.5114/reum.2015.57639. Epub 2016 Feb 11. PMID: 27407266; PMCID: PMC4847283.
Response: We really appreciate your kind suggestion and cite the reference on the main text.
4. In some places the use of English could be improved on. Completing this gaps will have an impact on the understanding the aim of the study and, from my point of view, is absolutely necessary.
Response: We asked a native speaker to correct the grammatical errors through our manuscript. We added his name Manini Daudi Romani on the author listed of our manuscript.